# Trained Innate Immunity in Animal Models of Cardiovascular Diseases

**DOI:** 10.3390/ijms25042312

**Published:** 2024-02-15

**Authors:** Patricia Kleimann, Lisa-Marie Irschfeld, Maria Grandoch, Ulrich Flögel, Sebastian Temme

**Affiliations:** 1Institute of Molecular Cardiology, Faculty of Medicine, University Hospital, Heinrich-Heine-University, 40225 Düsseldorf, Germany; patricia.kleimann@uni-duesseldorf.de (P.K.); floegel@uni-duesseldorf.de (U.F.); 2Department of Radiation Oncology, Faculty of Medicine, University Hospital, Heinrich-Heine-University, 40225 Düsseldorf, Germany; lisa-marie.irschfeld@med.uni-duesseldorf.de; 3Institute of Translational Pharmacology, Faculty of Medicine, University Hospital, Heinrich-Heine-University, 40225 Düsseldorf, Germany; maria.grandoch@uni-duesseldorf.de; 4Cardiovascular Research Institute Düsseldorf (CARID), University Hospital, 40225 Düsseldorf, Germany; 5Department of Anesthesiology, Faculty of Medicine, University Hospital, Heinrich-Heine-University, 40225 Düsseldorf, Germany

**Keywords:** trained innate immunity, cardiovascular diseases, monocytes, bone marrow, animal models

## Abstract

Acquisition of immunological memory is an important evolutionary strategy that evolved to protect the host from repetitive challenges from infectious agents. It was believed for a long time that memory formation exclusively occurs in the adaptive part of the immune system with the formation of highly specific memory T cells and B cells. In the past 10–15 years, it has become clear that innate immune cells, such as monocytes, natural killer cells, or neutrophil granulocytes, also have the ability to generate some kind of memory. After the exposure of innate immune cells to certain stimuli, these cells develop an enhanced secondary response with increased cytokine secretion even after an encounter with an unrelated stimulus. This phenomenon has been termed trained innate immunity (TI) and is associated with epigenetic modifications (histone methylation, acetylation) and metabolic alterations (elevated glycolysis, lactate production). TI has been observed in tissue-resident or circulating immune cells but also in bone marrow progenitors. Risk-factors for cardiovascular diseases (CVDs) which are associated with low-grade inflammation, such as hyperglycemia, obesity, or high salt, can also induce TI with a profound impact on the development and progression of CVDs. In this review, we briefly describe basic mechanisms of TI and summarize animal studies which specifically focus on TI in the context of CVDs.

## 1. A Brief Introduction into Trained Innate Immunity

Immune responses after sterile traumatic or ischemic tissue injury and against invading pathogens are very complex processes that aim to eliminate the injurious trigger and to restore the integrity and the functionality of the tissue. In general, immune responses can be separated into an innate and adaptive arm. Innate responses are initiated very rapidly within minutes and can last for hours or a few days. They are, for example, triggered by the detection of pathogenic microorganisms via pattern recognition receptors (PRRs). These PRRs recognize evolutionary conserved structures from classes of microorganisms, such as lipopolysaccharides of Gram-negative bacteria, or double-stranded RNA of certain viruses. However, PRRs can also be activated by endogenous molecules which are released after tissue damage [1]. Whereas the innate immune response is quite rapid and has a more or less broad specificity, the generation of a primary adaptive immune response requires more time. However, adaptive immune responses are highly specific against the pathogens or molecules that trigger the immune response. These rapid and highly specific immune responses are mediated by T cell and B cell clones that have been generated by selection and clonal expansion.

T cells recognize antigenic peptides which are presented via cell surface peptide receptors called MHC (major histocompatibility complex) molecules. MHC molecules are encoded within the *Mhc* locus that can be subdivided into class I, class II, and class III regions [2]. MHC class I (MHC-I) and class II (MHC-II) molecules are closely related in structure and function and both present antigenic peptides on the cell surface that can be recognized by the T cell receptor (TCR) of CD4^+^ or CD8^+^ T cells. Molecules of the MHC class III region are more heterogeneous and do not present peptides to T cells, but many of them are involved in immunological processes. MHC-I molecules are found on most nucleated cells and mainly bind cytosolic peptides that are generated by proteolysis via the proteasome and which are then transported into the endoplasmic reticulum. Within the endoplasmic reticulum (ER), these peptides are loaded onto MHC-I molecules. Peptide-loaded MHC-I molecules are then shuttled to the cell surface where they can be recognized by matching CD8^+^ T cells. In contrast, MHC-II molecules are mainly expressed by professional antigen-presenting cells (dendritic cells, macrophages, B cells) and are predominantly loaded with antigenic peptides that were derived from proteins located in the extracellular space. These proteins are internalized into the endolysosomal system by endocytosis or phagocytosis where they are degraded into small peptides by lysosomal enzymes. After transport of these MHC-II/peptide complexes to the cell surface, they can be identified by CD4^+^ T cells that express the corresponding TCR. However, it should be noted that autophagy can deliver cytosolic proteins into the late endosomal system that results in MHC-II presentation of cytosolic antigens [3]. Furthermore, cross-presentation describes the presentation of protein antigens that were taken up by phagocytosis/endocytosis by MHC-I molecules [4].

Activation of naïve CD4^+^/CD8^+^ T cells via peptide-loaded MHC-I/-II molecules results in proliferation and differentiation into multiple different subtypes. Although the main function of CD8^+^ T cells is to kill pathogen-infected cells or eliminate tumor cells, there are also CD8^+^ T cells with immunomodulatory or even suppressive function [5]. CD4^+^ T cells display an even larger amount of variety. Classically, helper CD4^+^ T cells were divided into T_H_1 or T_H_2 subtypes, depending on the secreted cytokine profile and their effector functions. T_H_1 cells promote cellular immune responses (e.g., activation of macrophages), whereas T_H_2 cells enhance antibody production. In recent years multiple additional CD4^+^ subtypes, such as T_H_17, T_H_9, T_H_22, and regulatory T cells (T_reg_), or follicular helper T cells (T_FH_), were identified. T_H_17 cells are involved in many autoimmune diseases [6], regulatory T cells can balance excessive immune responses and prevent autoimmunity [7], and CD4^+^ T_FH_ support B cell responses in the germinal centers of lymph nodes or the spleen [8].

The main function of B cells is to secrete antibody molecules that can opsonize and/or neutralize viruses, bacterial pathogens, or toxins. Furthermore, antibodies that bind to the surface of virus-infected cells or tumor cells can also be recognized by natural killer (NK) cells that result in the elimination of these cells by a process called antibody-dependent cytotoxicity. Activation of B cells involves the recognition of foreign antigens via cell surface expressed B cell receptors that result in internalization, processing of the antigens, and loading on MHC-II molecules. CD4^+^ T cells that possess a matching TCR recognize these MHC-II peptide complexes on the cell surface of the B cells which results in bidirectional activation. This process also induces the maturation of B cells and the secretion of large amounts of antibody molecules.

Innate and adaptive immune responses are not separate processes, but intensively work together to eliminate invading pathogens, to neutralize injurious agents, counteract tissue damage, and to restore tissue integrity. Dendritic cells (DCs) nicely show the crosstalk of innate and adaptive immunity [9]. Under inflammatory conditions, immature DCs internalize enormous amounts of local antigens and are, in parallel, activated by danger- or pathogen-associated molecular patterns. This induces the maturation of DCs, which is associated with strong upregulation of MHC-mediated antigen presentation and expression of costimulatory molecules, as well as their migration into draining lymph nodes. Here, mature DCs are scanned by CD4^+^/CD8^+^ T cells for matching peptide-loaded MHC-I/-II molecules, which finally results in the activation of appropriate T cells.

It has been believed for long times that the innate immune system always acts in an identical, nonspecific way. This has been challenged in the past decade as evidence accumulated that innate immune cells, such as monocytes, macrophages, or natural killer (NK) cells display long-term enhanced responses upon restimulation, which has been termed “trained innate immunity” (TI) [10]. However, there are several differences compared to the memory formation of the adaptive immune system. (i) After a first encounter with a stimulus that induces an innate training effect, the increased secondary response is not limited to the initial trigger but displays a rather general enhanced response to exogenous and endogenous stimuli [11]. (ii) The enhanced responsiveness is maintained via epigenetic and metabolic alterations in circulating immune cells, tissue resident cells, or bone marrow stem and progenitor cells [12]. (iii) Compared to the memory formation of the adaptive immune system that can result in lifelong resistance after natural infection or vaccination, the time period of this innate training is shorter and ranges from weeks to several months [10]. Nevertheless, stimulation of trained innate immune cells leads to a more rapid and stronger immune response with the secretion of higher levels of inflammatory cytokines, which can be either protective against infectious diseases or aggravate cardiovascular diseases (CVDs), for example atherosclerosis [13].

## 2. Stimuli and Mechanisms That Mediate Trained Innate Immunity

In this review, we briefly describe the main stimuli and some basic mechanisms of trained innate immunity. More detailed extensive descriptions are out of the scope of this review and can be found in several excellent reviews [10,11,14,15]. Initial work on trained innate immunity focused on microorganisms, microbial products including *Candida albicans* and its cell wall component β-glucan, as well as the Bacille Calmette–Guérin (BCG) vaccine [11], which revealed that innate training by infections or even vaccination can generate unspecific cross-protection (see Appendix A for an overview about some pathogens [16,17,18,19,20,21,22,23] and vaccines that induce TI [24,25,26,27,28,29,30,31]). However, now it is known that in addition to β-glucan and BCG, also chitin, muramyl-dipeptide, lipopolysaccharide (LPS), and self-derived molecules, such as uric acid, oxidized low-density lipoprotein (oxLDL), catecholamines, and hormones, like aldosterone, can also induce training of innate immune cells [11]. Even physical exercise [32] or *Plasmodium falciparum* [33] have been shown to be able to induce innate training of immune cells. Table 1 provides an overview about selected stimuli on human monocytes and their impact on trained innate immunity. Bacterial LPS is quite interesting because it has been shown that stimulation with low doses results in TI, whereas high doses of LPS lead to tolerance [34], indicating that long-term tolerance can also be a cause of innate stimulation.

Training of innate immune cells, like monocytes, macrophages, or NK cells, with β-glucan or oxLDL can lead to epigenetic alterations and/or metabolic adaptations that result in an enhanced response with elevated levels of cytokine release when these cells are stimulated a second time with an unrelated stimulus (see Figure 1 for a schematic overview). Classically, cultivated monocytes are stimulated with β-glucan and then these cells are restimulated after one week with a low dose of LPS, which leads to increased release of cytokines, such as interleukin-6 (IL-6), tumor necrosis factor-α (TNFα), or IL-1β [42]. It was shown that β-glucan-dependent TI is mediated by dectin-1 and the Raf-1 pathway, which results in changes in histone methylation at H3K4, like H3K4me1, H3K4me3, or H3K27ac [43]. These epigenetic alterations lead to an open chromatin conformation at promoters and enhancers of proinflammatory cytokines, like TNFα, IL-6 or interferon-γ (IFNγ).

TI also leads to metabolic alterations that are in part mediated by epigenetic mechanisms, or vice versa. Training of monocytes and macrophages with β-glucan induces a metabolic switch to aerobic glycolysis, also called the Warburg effect [44]. TI-induced histone modifications can lead to an upregulation of glycolytic enzymes that result in increased glucose consumption, lower basal respiration rate, production of lactate, and increased ratio of NAD^+^ (nicotinamide adenine dinucleotide) to NADH [45]. These metabolic adaptations are mediated by the mTOR/HIF-1α (mammalian target of rapamycin/hypoxia inducible factor-1α)pathway, as, for example, shown in mice where treatment with the mTOR inhibitor metformin and the myeloid specific knockout of HIF-1α abolishes trained immunity [46]. Besides, metabolic alterations, such as an increase in fumarate or acetyl-CoA, can directly trigger epigenetic modifications, such as histone acetylation and methylation [14].

Furthermore, in addition to various in vitro or ex vivo experiments with different stimuli, effects of trained innate immunity were also shown in multiple animal studies (see Section 4 and Section 5, Appendix A) and humans, for example after BCG vaccination [24] or already in hepatitis B (HBV) virus exposure in utero [20].

## 3. Trained Innate Immunity and Cardiovascular Diseases

Stimulation or training of the innate immune system can be a double-edged sword. It can either support the protection against infectious microorganisms [10,47,48] or enhance cardiovascular or chronic kidney diseases [13,49,50,51,52,53]. Furthermore, cardiovascular risk factors, such as an unhealthy diet, hypercholesterolemia, hyperglycemia, smoking, and stress, influence the innate immune system, particularly hematopoietic progenitor cells in the bone marrow. This leads to functional changes in these cells that support chronic metabolic and vascular inflammation and, therefore, can promote CVDs [49,50,52]. These cardiovascular risk factors are often associated with low-grade inflammation which—as already mentioned before—can induce TI, but may also be maintained and aggravated by TI, leading to a chronic non-resolving state. However, the precise interaction of cardiovascular risk factors with low-grade non-resolving inflammation, TI, and its influence on the development and progression of certain CVDs is largely unclear.

The largest body of evidence for a direct connection between TI and CVDs exists for atherosclerosis [13]. Oxidized LDL is a key factor for the development and progression of atherosclerosis, for example by triggering the growth of foam cells and sustaining a proinflammatory milieu within the arterial wall [51]. However, oxLDL has also been shown to induce trained innate immunity in human monocytes and macrophages by inducing a metabolic switch to glycolysis. This effect depends on mTOR signaling and epigenetic reprogramming, as observed through enriched histone mark H3K4me3 on genes encoding various proatherogenic cytokines and chemokines [35,42,54]. Furthermore, oxLDL induces an increase in scavenger receptors and a reduction in cholesterol efflux transporters, which is associated with enhanced foam cell formation [35]. Therefore, it is considered that oxLDL-induced trained immunity may be part of a vicious circle that could aggravate the development and progression of atherosclerosis.

The connection between TI and CVDs is further supported by ex vivo and in vitro analyses of cells obtained from human patients. Monocytes from patients with symptomatic coronary artery disease showed enhanced cytokine production capacity after differentiation into macrophages for 5 days [55]. Additionally, the authors observed that the reactive oxygen species (ROS)—which is generated in mitochondria—is regulated by the glycolytic enzyme pyruvate kinase M2 (PKM2). PKM2 boosts the production of the proinflammatory cytokines IL-6 and IL-1β via the transcription factor STAT3 (signal transducers and activators of transcription) which is again characteristic for TI. In addition to atherosclerosis or coronary artery disease, TI was also observed in perivascular adipose tissue (PVAT) of abdominal aortic aneurysms (AAAs) [56]. PVAT was of interest because its dysfunction affects large arteries primarily through immune cell infiltration. There is evidence that both innate and adaptive immune cells present in the PVAT of AAAs contribute to the maintenance of an injurious circuit, and TI may play a role in this crosstalk. Reverse engineering analysis was used to associate subsets of differential expression genes with cell type-specific histone modifications. Finally, they observed that dilated PVAT of small and large AAAs was associated with, e.g., H3K4me3/H3K4me1/H3K27ac signatures in CD14^+^ monocytes, which is typical for TI [56].

We have mentioned the detrimental effects of TI in connection with CVDs and specifically with oxLDL, but there are also stimuli that have the ability to suppress oxLDL-induced TI [57]. Changes in the balance of intracellular steroid hormones were observed in oxLDL-stimulated monocytes. Further analyses showed the unique ability of progesterone to attenuate the enhanced TNFα and IL-6 production via the nuclear glucocorticoid and mineralocorticoid receptors following oxLDL-induced TI [57]. This could be an interesting target to suppress the development of atherosclerotic plaques.

Overall, it can be stated that in almost all human studies, TI was associated with a detrimental effect on CVDs, which was mainly characterized by epigenetic changes and the resulting increased production of proinflammatory cytokines which might lead to aggravation of cardiac and vascular diseases (see Figure 2 for an overview about the complexity that interconnects TI with CVDs). But it is extremely difficult to specifically unravel the impact of TI on CVDs because TI itself is variable and, additionally, CVDs are very complex diseases that are affected by numerous cellular, physiological, and genetic variables.

## 4. Interconnections between Trained Immunity and Adaptive Immune Responses

One aspect that has not been intensively investigated so far is the influence of trained immunity on adaptive immune responses. The potential crosstalk between TI and T cell immunity was recently addressed in a very interesting review article [58]. We performed a literature search for “B cells and trained immunity”, but we did not find matching hits, further emphasizing that this field is relatively unexplored.

It is plausible that TI somehow impacts on T cell and B cell responses because it can alter the local micromilieu in which the cells become activated and differentiate into subtypes. Evidence for an interaction between TI and T cell responses was found in animals subjected to β-glucan stimulation, where splenic macrophages showed elevated levels of the costimulatory molecule CD80 [59]. Another study revealed that alveolar macrophages that were primed by T cells during a respiratory infection had higher levels of MHC class II molecules [60]. Both studies indicate that TI might lower the threshold for T cell activation. The expression of MHC-I and -II molecules is modified by several cytokines that are involved in TI, such as TNFα or IFNγ. Recently, it has been found that *Bacillus subtilis* spores in combination with influenza inactivated virus leads to innate training of dendritic cells which increases the generation of tissue resident memory T cells [61]. Of note, there is also evidence that adaptive immune responses have the capability to enhance the training state of cells which increase secondary responses to an unrelated stimulus. This phenomenon was found by Yao et al., who showed that the contact between CD8^+^ T cells with alveolar macrophages induces a memory phenotype in these macrophages that results in enhanced immune response against an infection with *Streptococcus pneumoniae* [60].

Another interesting question is if adaptive immune cells can also be trained by innate stimuli which result in epigenetic and metabolic changes that influence their functionality during adaptive immune responses. Administration of high dietary salt can induce a state of trained immunity with impaired healing of brain injury [62] (see also Section 5.3). High concentrations of sodium can increase blood pressure, but it can also enhance the proinflammatory activity of T cells and their differentiation into the T_H_17 lineage [63]. In addition, high sodium concentrations impair the functionality of T_reg_ responses [64]. In this context, it has been shown that repeated hypertensive stimuli with the nitric oxide synthase inhibitor L-NAME (N(ω)-nitro-L-arginine methyl ester) followed by a washout period and high salt treatment resulted in high numbers of CD4^+^ and CD8^+^ memory T cells within the kidney and the bone marrow [65]. However, whether or not high salt can induce epigenetic and metabolic alterations that result in an enhanced adaptive response is unknown.

There is a large body of evidence that T cell and B cell responses impact on cardiovascular diseases [66,67,68] and that there is a crosstalk between innate as well as adaptive immune responses during their initiation and progression [69,70]. If and how trained innate immunity engages in this process is currently also unclear.

## 5. Trained Innate Immunity in Animal Models of Cardiovascular Diseases

As already mentioned above, there is epidemiological, experimental, and conceptional evidence that trained innate immunity has a significant impact on CVDs. Due to the huge complexity and the interaction of different organs and multiple cell types, well-designed animal studies are especially important to unravel the relevance of TI on CVDs. However, to our knowledge, the number of animal studies that specifically focus on effects of trained immunity for the development and progression of cardiovascular diseases is relatively limited. A systematic search in PubMed for studies that investigated the impact of TI in animal models of CVDs with various combinations of key words revealed about 100 results for TI and CVD in general. From these, ~10 results were about animal studies that dealt with trained innate immunity in the context of CVDs.

In the following section we summarize and discuss some of these studies in detail and in particular extract the information that supports the trained immunity effect. Most of the studies investigated the relevance of TI for the development and progression of atherosclerosis, while others provide evidence of TI for stroke or in the context of abdominal aortic aneurysms (see also Table 2 for a comprehensive overview). At the end of this chapter, we also summarize one very interesting example of a contradictory finding where myocardial infarction leads to long-term reprogramming that results in immune suppression rather than activation.

**Table 2 ijms-25-02312-t002:** Animal studies of trained innate immunity in the context of cardiovascular diseases.

Species	Disease/Effect	TI Trigger	2nd Stimulus	Biological Effect/Mechanism	Evidence for Trained Immunity	References
Mouse (LDLr-/-)	Atherosclerosis (increased)	Hypercholesterolemia (LDLr-/-)	High-fat diet (HFD)	Activation and differentiation of bone marrow progenitors. Functionality of macrophages.	Indirect. Adoptive transfer of LPS-primed macrophages.	[71]
Mouse (LDLr-/-)	Atherosclerosis (increased)	Hypercholesterolemia (LDLr-/-)	Western diet (WD)	Bone marrow cells and splenocytes isolated after four weeks of WD followed by four weeks on a normal chow diet showed increased release of cytokines and chemokines after stimulation. Mechanism: IL-1 and NLRP3 inflammasome.	Direct. 4 weeks WD follwed by 4 weeks normal chow diet: epigentic alterations in bone marrow progenitors and increased response after restimulation with TLR ligands.	[72]
Mouse (C57BL/6 and LDLr-/-)	Atherosclerosis (increased)	Hyperglycemia (streptoztocin-induced diabetes)	Western diet (WD)	Bone marrow-derived macro-phages: enhanced M1 polarization and foam cell formation. Transfer of bone marrow from diabetic mice enhanced athero-sclerosis in LDLr-/- mice	Direct. Epigenetic modifications (H3K4me3 and H3K27ac), in bone marrow progenitor cells. Enhanced glucose uptake and lactate-production was main-tained after differentiation into macrophages.	[73]
Mouse (ApoE-/-)	Atherosclerosis (increased)	Super low-dose lipopolysaccharide (5 ng/kg BW)	Western, high-fat diet (HFD)	Low-dose LPS and high-fat diet, led to an increased pro-inflammatory situation and aggravated atherosclerosis. Proinflammatory macrophages reduce IRAK-M by degradation of SMAD-4 that results in the generation of non-resolving inflammatory monocytes.	Indirect. 4 week LPS + HFD, followed by 4 weeks HFD aggravated athero-sclerosis.	[74]
Mouse (ApoE-/-)	Abdominal aortic aneurysms	High-fat diet (HFD)	Angiotensin II (Ang II)	RNAseq analyses: HFD and AngII had a stronger effect on metabolic reprogramming and inflammation in the abdominal aorta compared to the thoracic aorta.	Indirect. Upregulation of genes involved in trained innate immunity.	[75,76]
Mouse	Stroke (increased)	High salt (HS)	Stroke	High-salt treatment impaired the resolution of the intracranial hematoma. and polarized macrophages to a less reparative phenotype. Downregulation of NR4a1 in macrophages mediates the HS effects on stroke recovery.	Direct. Reprogramming of hematopoetic progenitor cells (glycolysis, fatty acid metabolism, oxidative phosphorylation). Transfer of bone marrow from HS-treated mice.	[62]
Mouse	Cancer	Breast cancer (E0771 cells)	Myocardial infarction (MI)	MI increased the growth of the tumor. MI induces the release of Ly6chi monocytes with a suppressor phenotype that infiltrate the tumor and diminish the tumor response (CD8^+^ T cells, T_regs_).	Direct. Epigenetic alterations in bone marrow progenitor cells and tumor macrophages—reduced accessibility of genes for activation and cytokine production. Bone marrow transfer of mice with MI and cancer to naive mice enhanced cancer growth.	[77]

### 5.1. Atherosclerosis

Atherosclerosis is a chronic inflammatory disease that results in the formation of lipid-rich plaques within the arterial wall that contain high amounts of inflammatory cells, in particular macrophages. One of the main risk factors for atherosclerosis is hypercholesterolemia, which is characterized by elevated levels of circulating low-density and very-low-density lipoproteins (LDL, VLDL) and decreased levels of high-density lipoproteins (HDL). Disease progression is also strongly associated with inflammatory processes, particularly involving monocytes and macrophages [78]. Of note, there is a close link between hypercholesterolemia and inflammation, as it can directly promote the generation and release of monocytes and neutrophils from the bone marrow into the blood stream [79]. Whereas HDL reduces the proliferation of bone marrow progenitors, LDL increases their proliferation and promotes the differentiation of these cells into monocytes and granulocytes [80]. In this context, Seijkens et al. [71] observed that hypercholesterolemia in LDL receptor-deficient mice (LDLr-/-) induced the activation and differentiation of bone marrow progenitor cells which led to an increased plaque size with a more mature phenotype that contained elevated levels of macrophages and neutrophils. This study already showed that bone marrow progenitor cells can have an impact on atherosclerosis, although the precise mechanistic details, such as the epigenetic/metabolic adaptations of the innate immune system, have not been unraveled.

Cellular uptake of LDL does not result in massive cholesterol accumulation, because intracellular cholesterol downregulates the LDL receptor (LDLr). However, oxidation of LDL in serum or the vascular wall by enzymatic or non-enzymatic mechanisms (lipoxygenases, myeloperoxidases, metal ions) results in the generation of oxidized LDL variants (oxLDL) that do not bind to the LDLr but are internalized by scavenger receptors [81,82]. Scavenger receptors, such as SR-A and CD36, are not downregulated by oxLDL, and mediate massive cholesterol uptake that results in the formation of foam cells [81]. Cell-culture studies with human monocytes revealed that oxidized LDL can also induce epigenetic modifications that generate innate memory and result in increased cytokine release after restimulation with toll-like receptor 2 (TLR2) or TLR4 agonists [35]. Therefore, it was reasonable to speculate that if mice were fed with a Western diet (WD), they would also develop a training of the innate immune system that can impact on atherosclerosis development and progression [72]. Treatment of LDLr-/- mice (female, 8 weeks of age) with a Western diet (17.3% protein, 21.2% fat, 48.5% carbohydrates) for four weeks resulted in increased circulating cholesterol levels as well as growth factors, chemokines, and cytokines, which are indicators for a systemic inflammatory situation. These proinflammatory mediators were retained to normal after an additional period of four weeks with a normal chow diet (CD). However, when bone marrow cells or CD11b^+^ cells from the spleen isolated from mice treated with WD followed by a normal CD for four weeks were restimulated with several TLR ligands, the secretion of cytokines and chemokines was strongly elevated. This was associated with transcriptional and functional reprogramming of myeloid precursor cells. Furthermore, the authors revealed that IL-1 signaling and activation of the NLRP3 inflammasome are important for the WD-mediated trained immunity phenotype. NLRP3-/- and LDLr-/- double knockout mice treated with WD did not show an enhanced proliferation of bone marrow progenitors or increased numbers of circulating immune cells, and no there was no aggravated response to LPS-stimulation. Additionally, mice lacking NLRP3 and LDLr revealed markedly reduced atherosclerotic plaque size after feeding with a WD for eight weeks.

Another important risk factor for atherosclerosis is diabetes mellitus, and several pathophysiological mechanisms are considered to drive atherosclerosis in diabetic patients, such as dyslipidemia, increased levels of LDL, oxidative stress, hyperglycemia, and increased inflammation [83]. One main characteristic of diabetes is chronic hyperglycemia caused by a defect in insulin production and secretion, insulin action, or both. To investigate the connection between hyperglycemia, trained immunity in bone marrow progenitor cells, and atherosclerosis, Edgar et al. [73] used a mouse model of streptozotocin-induced diabetes in wild-type C57BL/6 mice (12–14 weeks old). After six weeks, bone marrow-derived macrophages of these diabetic mice displayed enhanced M1-associated IL-6 gene expression. After treatment of these cells with IL-4, they also showed lower levels of M2-markers (Ym1 and Fizz1). Furthermore, these cells also showed higher levels of modified LDL uptake and foam cell formation. Experimental evidence that hyperglycemia mediated TI does increase atherosclerosis was obtained by the transfer of bone marrow (BM) from diabetic or control mice into LDLr-/- mice, which are prone to the development of atherosclerotic plaques upon WD. Mice that received diabetic BM had much higher levels of atherosclerosis than mice that were treated with BM from control mice. Moreover, hyperglycemia also led to characteristic epigenetic modifications of histones (H3K4me3 and H3K27ac) that were associated with higher glucose uptake and lactate production and that were maintained after differentiation into macrophages, which led to an increased proinflammatory and proatherogenic phenotype. This study nicely revealed that chronic hyperglycemia is sufficient to induce epigenetic modifications in bone marrow progenitor cells and that these modifications lead to an increased proinflammatory phenotype of bone marrow-derived immune cells that have the capability to aggravate atherosclerosis.

In recent years, it has been revealed that low-grade non-resolving inflammation, which is associated with other chronic diseases, such as obesity or diabetes, is a further risk factor for cardiovascular diseases. It has been speculated that obesity and diabetes induce leakiness of the gut which results in low levels of circulating lipopolysaccharide (LPS). LPS, which is part of the outer membrane of Gram-negative bacteria, is recognized predominantly by myeloid immune cells, like macrophages, monocytes, or neutrophils. However, many non-immune cells also express TLR4 and respond to LPS stimulation and the release of inflammatory mediators, like TNFα, IL-1β, or IL-6. Recognition of LPS is a complex process and involves the LPS-binding protein (LBP) which transfers an LPS monomer to CD14. CD14 then delivers the LPS molecule to MD-2 (myeloid-derived protein 2) that finally results in the formation of a trimeric LPS/MD-2/TLR4 complex [84].

Circulating plasma concentrations of LPS in endotoxemic animals or patients are relatively low (100 pg/mL^–1^ ng/mL) [85]. These low-grade LPS concentrations induce a sustained stimulation of the innate immune system, rather than the induction of tolerance. To investigate if low-grade inflammation can indeed aggravate atherosclerosis, Geng et al. used apolipoprotein E (ApoE)-deficient mice (ApoE-/-, Jackson, male, 7–10 weeks) that received a high-fat diet (HFD) and were additionally treated with 5 ng/kg LPS every 2–3 days for 4 weeks, followed by HFD only for additional 4 weeks. This resulted in increased plasma cholesterol, and elevated plasma levels of TNFα, IL-6, IL-10, and MCP-1 (monocyte chemoattractant protein-1), as well as higher numbers of atherosclerotic macrophages. To test if subclinical LPS-stimulation of macrophages can enhance atherosclerosis, in vitro LPS-stimulated bone marrow macrophages were then applied to HFD-treated ApoE-/- mice, which resulted in significantly increased plaque sizes. Mechanistically, it was revealed that low-dose LPS stimulation inhibits the expression of the serine/threonine kinase IRAK-M via the micro RNA-24 (miR-24)-triggered degradation of the transcription factor SMAD-4. Consequently, this leads to the development of non-resolving inflammatory macrophages that drive the growth of atherosclerosis. This study did not specifically focus on trained innate immunity. However, it is likely that low-dose LPS leads to metabolic and epigenetic adaptations in bone marrow progenitors and sustains a systemic proinflammatory milieu that aggravates atherosclerosis and other cardiovascular or autoimmune diseases. Interestingly, it has been shown that β-glucan can reverse LPS-induced tolerance [86], indicating that a trained innate or tolerant immune state displays some kind of plasticity. In this context, it would be interesting if the administration of a high dose of LPS after a chronic super-low-dose LPS induces tolerance and may ameliorate atherosclerosis.

### 5.2. Abdominal Aortic Aneurysms

Two important types of aortic diseases which are closely linked to atherosclerosis are abdominal aortic aneurysms (AAAs) and dissections (AADs), which both cause over 10 000 deaths in the United States each year [87]. The exact pathophysiological mechanisms that lead to the development of AAAs/AADs have not been fully resolved, but studies show that inflammatory processes and in particular monocytes and macrophages are critically involved in the pathogenesis of AAAs/AADs [88]. To our knowledge, up to now, no animal studies have been published that specifically investigated the impact of innate immune training on the development and progression of abdominal aortic aneurysms and dissection.

But two recent animal studies and data analyses of animal studies provide at least evidence that TI could also be involved in the development and progression of aortic aneurysms [75,76]. One study found that ER stress and TI might play a role for thoracic or abdominal aortic aneurysms [75]. For this, the authors utilized 9–10 weeks old apolipoprotein E-deficient (ApoE-/-) mice that were treated with angiotensin II (AngII, 1000 ng/kg/min) via aortic minipumps and in parallel fed with a HFD (2% cholesterol and 20% fat) for 28 days to foster the development of aortic aneurysms. After four weeks of treatment, the aortas were explanted, cut in two halves (thoracic and abdominal part) and subjected to RNA sequencing (RNAseq). The obtained datasets revealed that AngII stimulation bypassed HFD-induced metabolic reprogramming and induced strong inflammatory responses in the abdominal aorta that were not found in the thoracic aorta. This could be one additional reason for the fact that aneurysms mostly happen in the abdominal part of the aorta. Furthermore, transcriptomic analyses showed that different sections of the aorta have unique signatures in pathological conditions. The authors conclude from the RNAseq data that sensing of various DAMPs initiates ER stress that mediates AngII-accelerated TI and leads to a different susceptibility of the thoracic and abdominal aorta to diseases. In a previous publication, the same group performed a systematic data analysis of several published studies where gene expression and the secretome were analyzed in atherosclerotic mouse aortas, or aortas of mouse models of AngII- or elastase-mediated aortic aneurysms. With this approach, the authors wanted to gain insights into the role of the aorta as an immunological organ and the impact of secretory genes for different aortic diseases. In this study, the authors found that in particular early secretomes may function as drivers for TI [76]. As already mentioned above, these two studies do not provide strong experimental evidence that TI is a driver for the development and progression of aortic aneurysms, but at least they show that TI might be involved. Future studies are necessary to investigate if animals subjected to defined stimuli of trained immunity, like β-glucan or BCG, lead to the aggravated formation of aortic aneurysms/dissections.

### 5.3. Stroke

High intake of dietary sodium (HS) is a hallmark of Western societies, and epidemiological studies have shown that HS impacts the development and progression of cardiovascular diseases [89]. It has been known for a long time that HS intake is associated with elevated blood pressure. More recent studies also revealed that HS drives the immune system into a more proinflammatory state that has a negative impact on cardiovascular and autoimmune diseases [90,91]. Mechanistically, HS can enhance the proinflammatory activity of T cells, monocytes, and macrophages [92,93,94,95]. This can occur either directly via elevated concentrations of sodium or indirectly by alterations in the gut microbiome that both drive the development of T_H_17 cells [63,96]. In this context, Lin et al. [62] investigated in mice if dietary HS induces trained innate immunity and if this has any impact on the long-term outcome after experimental induction of intracerebral hemorrhage (ICH). ICH is a very severe form of stroke that is caused by a rupture of a blood vessel inside the brain that causes the formation of an intraparenchymal hematoma. The hematoma is resolved over time, particularly by M2 polarized brain macrophages [97]. Therefore, the authors hypothesized that the influence of HS on the functionality of monocytes and macrophages could impact the resolution and healing after stroke. To investigate these questions, mice (4–6 weeks of age) were treated with a high-sodium chow diet (8% NaCl) and 1% NaCl in tap water for between two and eight weeks. They found that HS treatment indeed impaired the resolution of the hematoma, increased neutron loss, and polarized monocyte-derived macrophages to a less reparative phenotype. HS was also associated with metabolic and transcriptomic alterations in the bone marrow. Adoptive transfer of bone marrow from HS-treated mice diminished the recovery of the brain and led to a reduced reparative macrophage phenotype. Finally, the authors also revealed that the orphan nuclear receptor 4A1 (NR4A1) plays a central role in the HS mediated trained immunity phenotype. NR4A1 is of interest, because it regulates cell proliferation, glucose metabolism, and inflammatory responses, and is involved in macrophage activation as well as differentiation [98,99]. Using mice with a myeloid knockout of NR4A1 (LysM^Cre^), Lin et al. [62] found that these mice had larger intracranial hematoma, delayed neurobehavioral recovery, and a lower amount of reparative monocyte-derived brain macrophages.

### 5.4. Myocardial Infarction as as Trigger for Trained Innate Immune Suppression

Whereas the studies described above deal with the concept that training of the innate immune system impacts on CVDs, it is also possible that a CVD itself may induce innate immune priming. Epidemiological investigations have shown that there is a correlation between cancer and cardiovascular diseases, as the incidence of CVDs is higher in cancer patients, and patients with cancer have a higher risk of developing CVDs [100]. The phenotype and functionality of tumor-associated macrophages (TAMs) are an important determinant for the immune response against the tumor. Although TAMs are extremely heterogeneous, TAMs can be roughly divided into M1 and M2 subtypes. Whereas M1 TAMs enhance the immune response against the tumor tissue, M2 polarized tumor macrophages promote the formation of metastases, inhibit T cell responses, or stimulate angiogenesis, which all together results in tumor growth [101].

Acute myocardial infarction (MI) leads to the release of cytokines and chemokines and other danger molecules. This leads to systemic activation and release of monocytes from the spleen and neutrophil granulocytes from the bone marrow [102]. Therefore, it was interesting to investigate if the MI-mediated systemic reprogramming of the immune system might also influence the immune response against cancer. In their study, Koelwyn et al. [77] used a mouse model of breast cancer by implanting syngeneic E0771 breast cancer cells into the mammary fat pat. Myocardial infarction was induced by ligation of the left anterior descending artery (LAD) three days later. One of their key-findings was that the tumor size and volume 20 days post-implantation was twice as large in those animals with myocardial infarction. Analysis of the underlying mechanisms revealed that MI induces epigenetic alterations in bone marrow progenitors that leads to an increased release of Ly6c^hi^ monocytes, which then infiltrate the tumor. However, these Ly6c^hi^ monocytes did not enhance the anti-tumor response but had a phenotype of myeloid-derived suppressor cells (mMDSCs) that diminished anti-tumor CD8^+^ cells and enhanced the generation of T_regs_ inside the tumor. The long-term trained innate immunity effects of MI were verified by profound epigenetic alterations in bone marrow Ly6c^hi^ monocytes that revealed reduced accessibility for genes implicated in lymphocyte activation or cytokine production. Importantly, these epigenetic alterations were also found in mMDSCs in tumors of mice with MI, which shows that the phenotype and functionality of bone marrow progenitors can affect the tumor immune response. In addition, transplantation of bone marrow from mice with cancer and MI into naïve mice that were recovered for 14 weeks after bone marrow transfer before implantation of E0771 cancer cells resulted in accelerated tumor growth. Finally, the authors verified their findings that MI accelerated breast cancer growth in the mammary specific polyomavirus middle T antigen overexpression mouse model (MMTV-PyMT) which spontaneously develops not only mammary tumors but also metastases in the lungs [103]. This study is of particular interest, because it provides a contradictory viewpoint, as it shows that inflammatory stimuli can also lead to long-term immune suppression. Furthermore, it provides a mechanistic explanation of how cardiovascular diseases and tumor tissue interact, and it emphasizes that these interconnections must be considered during the treatment.

## 6. Concluding Remarks

In the past 10–15 years, numerous studies have revealed that innate immune cells can generate some kind of memory by epigenetic and metabolic alterations, which has a severe impact on secondary stimuli. Initially it was shown that microbial products, like β-glucan, BCG, or LPS, induce trained innate immune memory, but now it is known that endogenous molecules can also induce epigenetic and metabolic alterations. These are oxLDL, hyperglycemia, hypercholesterolemia, aldosterone, or dietary intake of high salt [35,40,62,72,73]. All these stimuli lead to distinct alterations in the epigenetic and metabolic phenotype which might vary between cells and tissues. Induction of trained immunity can also involve the mevalonate pathway [39], which stimulates the insulin growth factor 1 receptor (IGF1-R) or glutaminolysis that generates an epigenetic program via fumarate [37]. In combination with genetic differences, disease and vaccination history, nutrition, stress, or age, this phenomenon adds another level of complexity for the understanding of the development and progression of cardiovascular diseases.

Originally it was observed in humans that monocytes isolated from BCG-vaccinated persons displayed elevated levels of cytokine release after stimulation with TLR ligands, which has led to the idea that TI also affects myeloid bone marrow progenitors. Multiple follow-up studies in mice and in humans have unraveled that hematopoietic progenitor cells display metabolic and epigenetic alterations which result in an increased differentiation into monocytes or neutrophils with a pro- or anti-inflammatory phenotype. Most of the studies in mice have analyzed cells isolated from the bone marrow of the femur or the tibia. However, hematopoietic active bone marrow is also found in other bones, like the brachium, the antebrachium, or the sternum. We have recently observed that in mice, most neutrophil granulocytes that are released from the bone marrow into the circulation after experimental myocardial infarction are derived from the femur and tibia and to a lesser extent from the brachium or antebrachium [104]. Depending on the type and location of the inflammatory lesion, cells might also be recruited from different bone marrow locations. After stroke or aseptic meningitis, neutrophils are recruited from the bone marrow of the skull rather than from the tibia [105]. These cells seem to take a shortcut and migrate through small channels directly from the skull into the dura. If and how progenitor cells in the bone marrow apart from tibia and femur can display a trained innate immune phenotype and if this impacts on diseases is currently unknown.

Another form of long-term innate training can occur via long-lived tissue resident immune cells, like certain macrophages. For a long time, it was believed that tissue macrophages are derived from blood monocytes, which infiltrate into the tissue and replenish tissue resident macrophages. Fate mapping and single cell technologies have revealed that many organs contain self-renewing tissue macrophages that are originally derived from the yolk sac or the fetal liver and have populated the organs during embryogenesis [106]. Innate immune memory has been shown for microglia that acquired epigenetic alterations after systemic LPS application, and that these alterations persisted for six months and were able to aggravate β-amyloidosis in a murine model of Alzheimer’s disease [107].

Of note, epigenetic and metabolic adaptations are not unique to immune cells, as it has been shown that epigenetic modifications with an impact on cellular functionality also occur in vascular smooth muscle cells, endothelial cells, fibroblasts, or even mesenchymal progenitors [108,109,110,111]. For example, El-Osta et al. [112] demonstrated that exposing cultured endothelial cells to transient high glucose levels in vitro led to sustained hyperinflammation due to epigenetic alterations and modified gene expression. Another potential direct influence of TI on the end stage of cardiovascular diseases could occur via coronary smooth muscle cells. It was demonstrated that BCG and oxLDL can induce a trained immunity effect in human coronary smooth muscle cells in vitro. Increased proinflammatory cytokine production was observed upon restimulation after one week, mediated in part by the mTOR-HIF-1α signaling pathway [113]. It is largely unexplored if and how innate immune reprogramming affects long-lived tissue resident immune and non-immune cells and how these factors impact on diseases.

There is a large body of evidence that physical exercise is beneficial for cardiovascular diseases as it can either prevent the development of CVDs or can also have a positive effect on the outcome of the disease [114]. Moderate exercise leads to epigenetic and metabolic alterations in murine bone marrow-derived macrophages. These macrophages had an M2 signature and displayed a decreased induction of NFκB upon LPS-stimulation [32]. Another study found that exercise can also protect from ischemia reperfusion injury in the liver due to metabolic reprogramming of Kupffer cells [115]. The precise contribution of exercise-induced innate epigenetic reprogramming is currently unknown. However, it seems that exercise has the capability to rewire epigenetics and metabolism and, therefore, might be an interesting non-pharmaceutical approach to tackle the vicious cycle induced by non-resolving chronic inflammation.

In summary, there is a large body of evidence that trained innate immunity has an impact on cardiovascular diseases. Future preclinical studies are required to further unravel the impact and distinct mechanisms of trained immunity on different cardiovascular diseases, such as myocardial infarction or abdominal aortic aneurysms and dissections. Here, it would also be of particular interest to unravel the impact of TI in individual immune and non-immune cells on the development and progression of CVDs. Trained innate immunity is characterized by extensive crosstalk between different organs, such as the bone marrow, the spleen, the circulatory system, and the diseased target organs. Therefore, sophisticated animal studies will be extremely helpful to further elucidate the crosstalk between trained innate immunity and cardiovascular diseases. This could be helpful to better understand the pathophysiology of human diseases and find novel targets for therapeutic approaches.

## Figures and Tables

**Figure 1 ijms-25-02312-f001:**
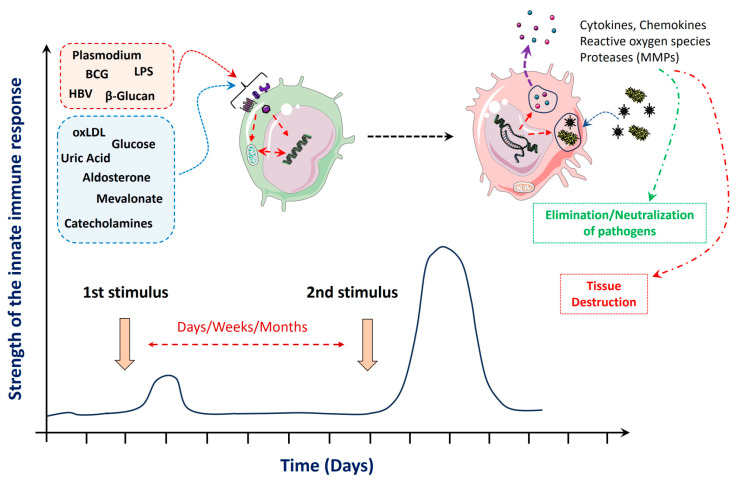
Stimuli and effects of innate immune training: Primary stimulation (1st stimulus) of innate immune cells (e.g., monocytes) by pathogen-derived or endogenous molecules leads to a transient immune response with elevated cytokine expression but can also induce metabolic and epigenetic adaptations that can be maintained over weeks or months. Stimulation of these cells a second time, even with an unrelated stimulus (this can occur after days, weeks or several months), results in an aggravated secondary response with enhanced release of inflammatory cytokines, chemokines, or reactive oxygen species. This can support the fight against infectious pathogens but can also lead to severe tissue destruction, in the context of autoimmune or autoinflammatory diseases. The curve in the lower part of the figure displays a schematic representation of an immune response of isolated monocytes stimulated in vitro with oxLDL (oxidized low-density lipoprotein) or β-glucan followed by a second stimulation (e.g., with lipopolysaccharide (LPS)) after one week. The x-axis displays the time (in days), and the y-axis represents the magnitude of the innate immune response. As shown by in vivo animal studies and ex vivo experiments with isolated cells from humans, an enhanced secondary response can also be found weeks or months after the first stimulus. Parts of the figure were drawn using pictures from Servier Medical Art. Servier Medical Art by Servier is licensed under a Creative Commons Attribution 3.0 Unported License (https://creativecommons.org/licenses/by/3.0/ (accessed on 7 January 2024)). (BCG = Bacille Calmette–Guérin; HBV = hepatitis B Virus; MMPs = matrix metalloproteinases).

**Figure 2 ijms-25-02312-f002:**
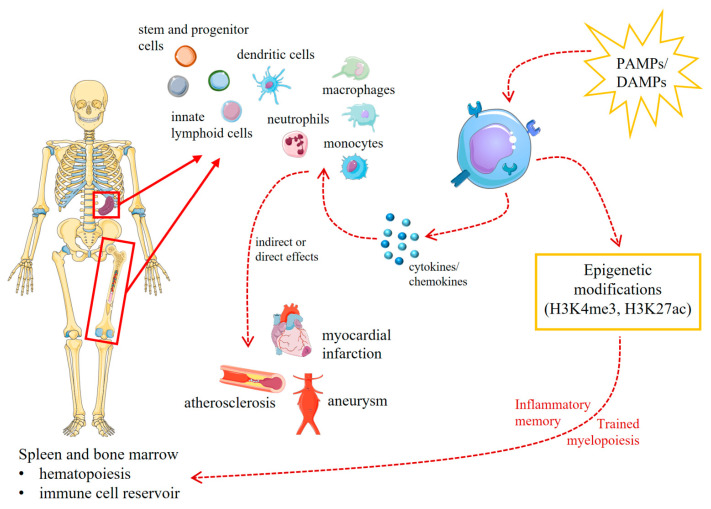
Schematic overview of the interactions between trained innate immunity and cardiovascular diseases. Various innate immune cell types including stem and progenitor cells that are mostly located in the spleen and bone marrow react directly or indirectly on different pathogen-associated molecular patterns (PAMPs) as well as damage-associated molecular patterns (DAMPs). This leads to an increased cytokine and chemokine production and activation of other immune cells. Furthermore, these stimuli can also induce epigenetic modifications and metabolic adaptations which cause innate inflammatory memory as well as trained myelopoiesis. These stimulated cells in turn have a direct or indirect effect on the development and progression of various CVDs, such as atherosclerosis, myocardial infarction, or aneurysms. Parts of the figure were drawn using pictures from Servier Medical Art. Servier Medical Art by Servier is licensed under a Creative Commons Attribution 3.0 Unported License (https://creativecommons.org/licenses/by/3.0/ (accessed on 7 January 2024)).

**Table 1 ijms-25-02312-t001:** Triggers for trained immunity in human monocytes and the biological effects.

Species	Cell Type	1st Stimulus	2nd Stimulus	Biological Effect	References
Human	Monocytes	oxLDL	TLR2 or TLR4 ligands	Long-lasting proatherogenic macrophage phenotype via epigenetic histone modifications Increased proinflammatory cytokine production and foam cell formation	[35]
Human	PBMCs	Uric acid	TLR2 or TLR4 ligands	Upregulation of proinflammatory cytokines	[36]
Human	Monocytes	Methylfumarate	LPS	HIF-1α degradation, histone methylation, and acetylation	[37]
Human	Monocytes	BCG	LPS	Increase in glycolysis and glutamine metabolism, regulated by epigenetic mechanisms at the level of chromatin organization	[38]
Human	Monocytes	Mevalonate	LPS	Increased function of IGF1 receptor and subsequent histone modifications in inflammatory pathways	[39]
Human	Monocytes	Aldosterone	TLR2 or TLR4 ligands	Proinflammatory cytokine production, ROS production, upregulation of fatty acid synthesis	[40]
Human	Monocytes	(Nor)adrenaline	LPS	Upregulation of proinflammatory cytokine production via the β-adrenoreceptor	[41]
Human	Monocytes	*Plasmodium falciparum*-infected red blood cells	LPS	Hyperproduction of IL-6 and TNFα	[33]

## Data Availability

No new data were created.

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
