# Peer review of "Trained Innate Immunity in Animal Models of Cardiovascular Diseases"

_ijms, 2024, doi:10.3390/ijms25042312_

Round 1

Reviewer 1 Report

Comments and Suggestions for Authors

The manuscript is devoted to regulation of innate imminity induced by different triggers and resulting in various pathologies. Description of epigenetic and metabolic mechanisms of the trained innate immunity and comparison with adaptive immune response is short, informative and clear. 

Line 41. "Innate responses occur very rapidly..."

Really, cyrokine gene expression differs. There are early and late cytokines. Probably, it would be reasonable to mention the range from a few minutes to days. 

Line 52. "MHC (major histocompatibility) molecules"

There are major histocompatibility complex of type 1 for Th1 immune response and MHC2 for Th2 response. Currently, after discovery of Th17 cells taking part in clearing extracellular pathogens and tissue inflammation, the paradigm of  the polariztion between Th1 and Th2  had been revised (Jiang, 2011). Besides 3 major Th1, Th2 and Th17 ways several minor T-helper subpopulations secreting certain sets of cytokines had been identified. It did not seem worth to mention general MHC without important details.

Fig. 1. Axis X - time without even approximate schematic scale. There is a big difference between the first cytokine gene expression maximum and subsequent trained innate immunity.  Axis Y is also without legend. What was detected RNA by means of RT2-PCR or proteins using ELISA or xMAP? 

Table 2: Trained innate immunity in animal studies of cardiovascular diseases.

Cancer does not belong to cardiovascular diseases. 

In the section 5 "Concluding Remarks" several contrapositions ("but") are without evident sense. The meaning of the whole sentences is not clear. 

Comments on the Quality of English Language

Minor mistakes are highlighted in yellow in the attached file. 

Abbreviation list is missing and some acronyms (such as NK, LDL and LDLs) remained without explanation in the text. 

Species of microorganisms should be written in Italic font. 

Author Response

#Reviewer 1

Comments and Suggestions for Authors

The manuscript is devoted to regulation of innate imminity induced by different triggers and resulting in various pathologies. Description of epigenetic and metabolic mechanisms of the trained innate immunity and comparison with adaptive immune response is short, informative and clear.

Response: Thank you very much for the positive evaluation of our manuscript!

Line 41. "Innate responses occur very rapidly..."

Really, cyrokine gene expression differs. There are early and late cytokines. Probably, it would be reasonable to mention the range from a few minutes to days.

Response: We fully agree and we have now added the time frame from minutes, hours to a few days. (Line 50)

Line 52. "MHC (major histocompatibility) molecules"

There are major histocompatibility complex of type 1 for Th1 immune response and MHC2 for Th2 response. Currently, after discovery of Th17 cells taking part in clearing extracellular pathogens and tissue inflammation, the paradigm of the polariztion between Th1 and Th2 had been revised (Jiang, 2011). Besides 3 major Th1, Th2 and Th17 ways several minor T-helper subpopulations secreting certain sets of cytokines had been identified. It did not seem worth to mention general MHC without important details.

Response: Thank you very much for being aware that this section was too superficial. Initially, we did not want to go into details, but we agree with your criticism. Therefore, we strongly extended this section to briefly introduce MHC class I and class II molecules, antigen processing and presentation and T cell subsets. (Lines 61-115)

Fig. 1. Axis X - time without even approximate schematic scale. There is a big difference between the first cytokine gene expression maximum and subsequent trained innate immunity. Axis Y is also without legend. What was detected RNA by means of RT2-PCR or proteins using ELISA or xMAP?

Response: We apologize that figure 1 might be misleading to the reader. The difference in the response peak after first/second stimulation was too big and we now adjusted the magnitude and the duration in revised Figure 1 (Lines 180-181).

However, we think it is difficult to add an appropriate scaling for the X- and Y-axis. The enhanced secondary response can be initiated after one week as shown by cell culture experiments, or after weeks/months as shown by animal studies or by ex vivo experiments with monocytes derived from BCG vaccinated persons. In our opinion a similar difficulty exists for the Y-axis depending on the readout (RNA, protein, functional assays e.g.). However, we tried to address these points by including a timescale for the X-axis (days) and we changed the title for the Y-axis. Furthermore, we strongly expanded the figure legend to explain that the curve mimics the initial in vitro experiments with isolated monocytes. We also included that in general the secondary response can occur after days, weeks or months. (Lines 182-199).

Table 2: Trained innate immunity in animal studies of cardiovascular diseases.

Cancer does not belong to cardiovascular diseases.

Response: We fully agree. Koelwyn et al. did not investigate the impact of trained innate immunity on the outcome of the MI. Furthermore, as addressed by reviewer 3, the study of Koelwyn is an example of a contradictory effect of innate immune training, because it results in the induction of myeloid derived suppressor cells. The reason why we included this very interesting study was that it shows how MI can induce alterations of the innate immune system that affect the course and outcome of other diseases. Furthermore, we changed the title of the table to better match it to the content. We hope that this study now fits to the studies listed in table 2. (Line 559).

In the section 5 "Concluding Remarks" several contrapositions ("but") are without evident sense. The meaning of the whole sentences is not clear.

Response: We apologize that we missed these mistakes. We have rewritten this part to enhance sense and clarity. (Lines 565-573)

Comments on the Quality of English Language

Minor mistakes are highlighted in yellow in the attached file.

Abbreviation list is missing and some acronyms (such as NK, LDL and LDLs) remained without explanation in the text.

Species of microorganisms should be written in Italic font.

Response: We severely apologize and corrected all of these mistakes. We did not include a separate list of abbreviations, but we carefully checked that all abbreviations and acronyms are explained.  

Reviewer 2 Report

Comments and Suggestions for Authors

Manuscript by Kleimann et al reviewed literature devoted to a relatively new scientific topic - trained innate immunity. This review focused especially on the available evidence from animal studies about the role of trained innate immunity in cardiovascular diseases. It would be beneficial adding to the review the description of the role of different parts of trained innate immunity (monocytes, neutrophils, dendritic cells, etc) in the pathogenesis of cardiovascular diseases. The authors should also consider a thorough proofreading of their manuscript for typos. 

Comments on the Quality of English Language

The manuscript is written well, only proofreading for typos is needed.

Author Response

#Reviewer 2

Comments and Suggestions for Authors

Manuscript by Kleimann et al reviewed literature devoted to a relatively new scientific topic - trained innate immunity. This review focused especially on the available evidence from animal studies about the role of trained innate immunity in cardiovascular diseases. It would be beneficial adding to the review the description of the role of different parts of trained innate immunity (monocytes, neutrophils, dendritic cells, etc) in the pathogenesis of cardiovascular diseases.

The authors should also consider a thorough proofreading of their manuscript for typos.

Response: Thank you very much for the evaluation of our manuscript. We agree that the role of trained innate immunity in combination with distinct immune cell subsets on the pathogenesis is a very important point. However, on the one hand we think that this very complex topic would be beyond the scope of our review and would be better covered by a separate review article. On the other hand – at least to our knowledge – it is currently relatively unclear how a trained phenotype of a specific immune cell subset influences the pathogenesis of a certain cardiovascular disease. Of course, this is one of the most important questions for future research in this field and therefore we included one sentence about this topic in the concluding remarks section (Lines 631-632)

Comments on the Quality of English Language

The manuscript is written well, only proofreading for typos is needed.

Response: We apologize and we have extensively and very carefully revised the whole manuscript to remove all errors and typos.

Reviewer 3 Report

Comments and Suggestions for Authors

-        The chosen topic of this scientific review paper is timely, relevant, and worth exploring, adding value to the existing body of knowledge in the field. The paper exhibits a well-structured format that facilitates clarity and ease of comprehension for the readers. The logical flow of information makes it easy to follow. Moreover, the depth of the review is notable, showcasing a comprehensive understanding of the subject matter.

-        I intend to enhance the paper further by providing comments on two major aspects of the review.

o   The correlation between TI and mild systemic inflammation remains ambiguous. The authors mention the possible involvement of „low grade non-resolving inflammation” only once. Should TI take place and experience a subsequent activation, it would logically lead to the delayed activation of adaptive immune cells. Consequently, TI is considered an integral component of immune activation rather than being solely responsible for it. Furthermore, B and T cells that have been activated can subsequently stimulate cells participating in TI. The paper does not address the impact of TI on adaptive immunity at this juncture.

o   The interpretation of the papers cited in Section 4.4 by the authors is inaccurate. According to Koelwyn et al.: „These findings suggest that MI-induced accelerated tumor growth is associated with an immunosuppressive intratumoral immune landscape.” Moreover, trained immunity-linked genes were found to be suppressed. These findings are contradictory to trained immunity. Neither ref. 68, nor ref.69 cover the topic of trained immunity. I recommend either removing this section or preserving it as an illustration of the opposing viewpoint.

-        Minor comments:

o   „Candida albicans” should be written in italics.

o   „plasmodium falciparum” should be written as Plasmodium falciparum

o   The mechanism of LDL oxidation should be described.

o   „low grade” should be written as low-grade

o   „The Phenotype” should be written as The Phenotype

Author Response

#Reviewer 3

 Comments and Suggestions for Authors

-        The chosen topic of this scientific review paper is timely, relevant, and worth exploring, adding value to the existing body of knowledge in the field. The paper exhibits a well-structured format that facilitates clarity and ease of comprehension for the readers. The logical flow of information makes it easy to follow. Moreover, the depth of the review is notable, showcasing a comprehensive understanding of the subject matter.

Response: Thank you very much for reviewing our manuscript and the constructive suggestions.

-        I intend to enhance the paper further by providing comments on two major aspects of the review.

o   The correlation between TI and mild systemic inflammation remains ambiguous. The authors mention the possible involvement of „low grade non-resolving inflammation” only once. Should TI take place and experience a subsequent activation, it would logically lead to the delayed activation of adaptive immune cells. Consequently, TI is considered an integral component of immune activation rather than being solely responsible for it. Furthermore, B and T cells that have been activated can subsequently stimulate cells participating in TI. The paper does not address the impact of TI on adaptive immunity at this juncture.

Response: Thank you very much for addressing these points. We agree that the connection between low-grade non-resolving inflammation could be emphasized. Therefore, we have now included low-grade inflammtion and TI into the abstract (Line 34) and we have rewritten a paragraph in chapter 3 to more clearly connect low-grade inflammation and TI. (Lines 214-219)

The second point about the connection between TI and adaptive immunity is very interesting. We do not fully agree that TI solely leads to delayed activation of adaptive responses. It could also be that TI might even enhance adaptive immune responses due to elevated activation of the immune cells, higher cytokine levels, elevated expression of MHC and costimulatory molecules. Additionally, TI might also alter the polarization of the responses. These aspects have been discussed in a recent review article by Murphy et al. (2021). However, although it is plausible that TI might impact on the activation and differentiation of adaptive immune cells, the knowledge in this field is relatively limited.

To cover the intersection of trained immunity and adaptive T cell responses, we have added a chapter on this topic (Lines 279-317).

o   The interpretation of the papers cited in Section 4.4 by the authors is inaccurate. According to Koelwyn et al.: „These findings suggest that MI-induced accelerated tumor growth is associated with an immunosuppressive intratumoral immune landscape.” Moreover, trained immunity-linked genes were found to be suppressed. These findings are contradictory to trained immunity. Neither ref. 68, nor ref.69 cover the topic of trained immunity. I recommend either removing this section or preserving it as an illustration of the opposing viewpoint.

Response: Many thanks for addressing this important point! The reviewer is perfectly right that the study of Koelwyn et al. is a nice example that inflammatory trigger can also lead to long term immunosuppressive effects. According to the suggestion of the reviewer we now present this study as an alternative outcome of innate immune stimulation. (Lines 516, 553-55)

-        Minor comments:

o   „Candida albicans” should be written in italics.

Response: Corrected

o   „plasmodium falciparum” should be written as Plasmodium falciparum

Response: Corrected

o   The mechanism of LDL oxidation should be described.

Response: We thank the reviewer for making us aware that this important information is missing. We now added a short paragraph that very briefly described LDL oxidation and why this is important for atherosclerosis. (Lines 358-364). 

o   „low grade” should be written as low-grade

Response: Corrected

o   „The Phenotype” should be written as The Phenotype

Response: Corrected